# Visual perception of liquids: Insights from deep neural networks

**Jan Jaap R. van Assen**[1]*, **Shin'ya Nishida**[1,2], **Roland W. Fleming**[3,4]

**1** Human Information Science Laboratory, NTT Communication Science Laboratories, Nippon Telegraph and Telephone Corporation, Atsugi, Kanagawa, Japan, **2** Graduate School of Informatics, Kyoto University, Yoshida-honmachi, Sakyo-ku, Kyoto, Japan, **3** Department of Experimental Psychology, University of Giessen, Giessen, Hessen, Germany, **4** Centre for Mind, Brain and Behaviour (CMBB), University of Marburg and Justus Liebig University, Giessen, Germany

* mail@janjaap.info

**Data Availability Statement:** All human data, statistical analysis code, stimuli, network training sets, and trained networks are available on Zenodo at link http://doi.org/10.5281/zenodo.3534568.

## Abstract

Visually inferring material properties is crucial for many tasks, yet poses significant computational challenges for biological vision. Liquids and gels are particularly challenging due to their extreme variability and complex behaviour. We reasoned that measuring and modelling viscosity perception is a useful case study for identifying general principles of complex visual inferences. In recent years, artificial Deep Neural Networks (DNNs) have yielded breakthroughs in challenging real-world vision tasks. However, to model human vision, the emphasis lies not on best possible performance, but on mimicking the specific pattern of successes and errors humans make. We trained a DNN to estimate the viscosity of liquids using 100.000 simulations depicting liquids with sixteen different viscosities interacting in ten different scenes (stirring, pouring, splashing, etc). We find that a shallow feedforward network trained for only 30 epochs predicts mean observer performance better than most individual observers. This is the first successful image-computable model of human viscosity perception. Further training improved accuracy, but predicted human perception less well. We analysed the network's features using representational similarity analysis (RSA) and a range of image descriptors (e.g. optic flow, colour saturation, GIST). This revealed clusters of units sensitive to specific classes of feature. We also find a distinct population of units that are poorly explained by hand-engineered features, but which are particularly important both for physical viscosity estimation, and for the specific pattern of human responses. The final layers represent many distinct stimulus characteristics—not just viscosity, which the network was trained on. Retraining the fully-connected layer with a reduced number of units achieves practically identical performance, but results in representations focused on viscosity, suggesting that network capacity is a crucial parameter determining whether artificial or biological neural networks use distributed vs. localized representations.

**Funding:** JJRvA was funded by the European Research Council (ERC) Consolidator Award 'SHAPE'–project number ERC-CoG-2015-682859 (https://erc.europa.eu), by Japan Society for the Promotion of Science (JSPS) KAKENHI Grant Number JP15H05915 (https://www.jsps.go.jp/english/), and a Google Faculty Research Award (https://ai.google/research/outreach/faculty-research-awards/). SN was funded by Japan Society for the Promotion of Science (JSPS) KAKENHI Grant Numbers JP15H05915 and 20H00603. RWF was funded by Deutsche Forschungsgemeinschaft (DFG, German Research Foundation, https://www.dfg.de/en/)–project number 222641018–SFB/ TRR 135 TP C1, by the European Research Council (ERC) Consolidator Award 'SHAPE'–project number ERC-CoG-2015-682859, and a Google Faculty Research Award. The funders had no role in study design, data collection and analysis, decision to publish, or preparation of the manuscript.

**Competing interests:** The authors have declared that no competing interests exist.

## Author summary

How the brain visually computes the physical properties of complex natural materials is a major open challenge in visual neuroscience. Here, we focussed on the perception of liquids—a particularly challenging class of materials due to their extreme mutability and diverse behaviours. We present the first image-computable model that can predict average human viscosity judgments from fluid simulation movies as well as individual observers can across a wide range of viewing conditions. We trained artificial neural networks to estimate viscosity from 100,000 20-frame simulations, and find that the models best predict human perception after relatively little training—long before they have reached optimal performance. This suggests that while human viscosity perception is remarkably good, even better performance is theoretically possible. Probing the networks with 'virtual electrophysiology' reveals many different features the networks use to estimate viscosity. Surprisingly, we find that the represented features are highly influenced by the size of the networks' parameter space, while prediction performance remains practically unchanged. This implies that some caution is required in making direct inferences between neural network models and the human visual system. However, the methods presented here provide a systematic framework for comparing humans to neural networks.

## Introduction

For centuries, researchers have tried to unravel the mechanics of the human visual system—a system that can successfully identify complex, naturalistic objects and materials across an unimaginably wide range of images. Many of the lower-level mechanisms within this system are now quite well understood [1–3]. For example, networks of cells have been identified that are specifically tuned to orientations, colours, spatial frequencies, temporal frequencies, motion directions, and disparities [4,5]. Cells further along the visual processing hierarchy are sensitive to more complex stimulus characteristics, and are much harder to characterize [6]. However, recent advances in artificial neural networks hold some promise for developing detailed, image-computable process models of sophisticated visual inferences, such as object recognition in arbitrary photographs [7–10].

Artificial neural networks provide an experimental platform for simulating complex visual abilities, and then carefully probing the role of specific objective functions, training sets and network architectures that yield human-like performance. By concentrating on a single task—such as the estimation of a particular physical property from the image—it becomes easier to single out the learned features of a network. Having developed a model that mimics human behaviour, the response properties of all units in the network can be measured with arbitrary precision over arbitrary conditions, like an idealised form of *in vivo* systems neuroscience performed on a model system rather than real tissue.

A particularly intriguing visual ability is the perception of liquids. Liquids can adopt an extraordinary range of different appearances because of their highly mutable shapes, which are influenced both by internal physical parameters, such as viscosity, and external forces, such as gravity. The most important physical property distinguishing different liquids is viscosity. Yet to estimate viscosity, the visual system must somehow discount the contributions of the external forces to the observed behaviour. For example, a viscous liquid can be made to flow and splash somewhat like a runny liquid if propelled with sufficient speed. The behaviour of liquids is governed by complex physical laws, and it is rather unlikely that we infer the viscosity of a given liquid by explicitly simulating the flow of particles within the liquid (although see

[11,12]). Previously, we found that observers draw on a range of optical, shape and motion cues to identify liquids and infer their properties [13–16]. However, the stimulus features underlying such inferences are often only loosely defined. To date there is still no image-computable model that can predict the perception of liquids or their viscosity. Here, we sought to leverage recent advances in deep neural networks (DNNs) to develop such a model and then probe its inner workings to generate novel hypotheses about how the human visual system estimates viscosity.

In machine learning, most work on artificial neural networks concentrates on achieving the best possible performance in a given task. In this study, by contrast, rather than seeking to develop a network that is mathematically optimal at estimating viscosity, instead we seek to develop a feedforward convolution network that most closely mimics the behaviour of the human visual system. To evaluate the extent to which models resembled humans, we asked observers to judge viscosity in the same movies that were shown to the trained neural networks.

The neural networks used here had a 'slow-fusion' architecture [17] for processing movie data (as opposed to static frames). They were trained on a dataset of 100.000 computer-generated fluid simulation animations, 20 frames long, depicting liquids interacting in ten different scene classes, which induced a wide variety of behaviours (pouring, stirring, sprinkling, etc; **Fig 1**). Their training objective was to estimate the physical viscosity parameter in the simulations. To test generalization, the tenth scene was not used during training and 0.8% of the simulations in each scene were withheld for validation during training. The training labels corresponded to the sixteen different physical viscosity steps that were simulated. For comparison, human observers performed a viscosity rating task, in which they viewed 800 of these stimuli and assigned perceived viscosity labels. The networks were trained on physical viscosity labels—not human ratings—but we used Bayesian optimization of the network's hyperparameters (e.g., learning rate, momentum) and layer specific settings (kernel sizes, number of filters) to search for networks that correlated well with humans on the 800 perceived viscosity labels. Importantly, training was relatively short with only 30 epochs (30 repetitions of the entire training set). With the networks in hand, we then analysed their internal representations to identify characteristics that led to human-like behaviour.

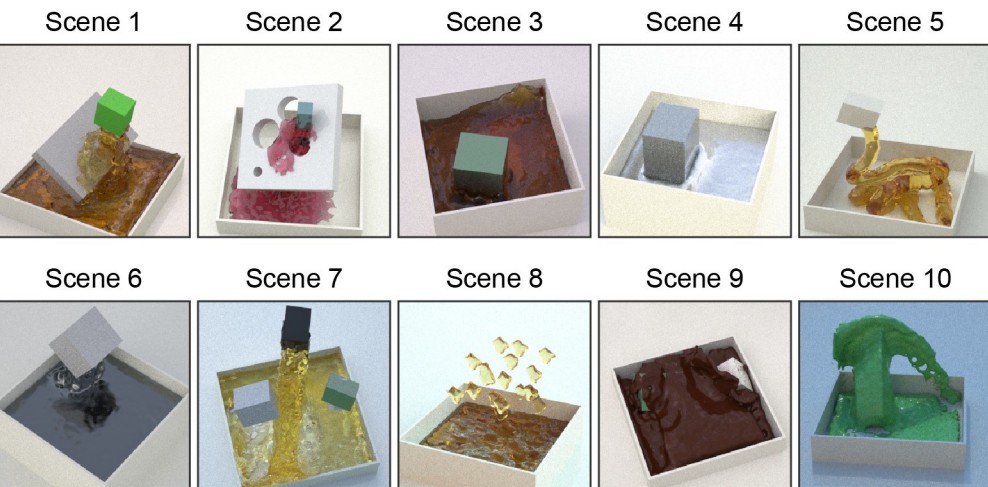

**Fig 1. Stimuli overview showing the ten different scenes.** Different liquid interactions were simulated, as pouring, rain, stirring and dipping. Optical material properties and illumination maps were randomly assigned with the white plane and square reservoir staying constant. **S1 Video** shows the moving stimuli.

Our main analyses and findings are as follows. To determine whether we have a model that is sufficiently close to human performance to warrant further analysis, we first compared the networks' predictions with human perceptual judgments on a stimulus-by-stimulus basis. We find that a network trained to estimate physical viscosity indeed predicts average human viscosity judgments roughly as well as individual humans do. This need not have been the case. Humans learn to perform a much wider range of visual tasks on a much more diverse visual diet, so it is not trivial that such a network trained on physical labels and computer simulations predicts both the errors and successes of human performance. We also find that the best predictions arise when networks are trained for a relatively short duration.

Second, having established that the network mimics human performance, we sought to gain insights into the inner workings of the network, by analysing the response properties of individual units at various stages of the network ('virtual electrophysiology'). We did this by: (a) comparing their responses to a set of hand-engineered features and ground-truth scene properties, (b) identifying stimuli that most strongly or weakly drive units, and (c) directly visualizing features through activation maximization. Together, these analyses revealed that many units are tuned to interpretable spatiotemporal and colour features. Yet we also find a distinct population of units with nontrivial responses properties (i.e., whose responses are poorly explained by any of the features we considered), and which are especially important for the performance of the network. We also show that linear combinations of the hand-engineered features are insufficient on their own to account for human viscosity perception, further reinforcing the importance of the additional units.

Third, we analysed network representations at the level of whole layers ('virtual fMRI'), and studied the effects of network capacity (i.e., number of units) on the internal representation. The main findings are: (1) a gradual transition from low-level image descriptors to higher level features along the network hierarchy, and (2) a striking dependency of the internal representations on the number of units, practically independently of overall performance and the ability to predict human judgments. This suggests that caution is required in inferring the properties of biological visual systems from models with seemingly similar performance.

Finally, we compared representations at the level of entire networks, to confirm whether 100 instances of the same architecture trained on the same dataset yielded similar internal representations ('virtual individual differences'). The results indeed reveal highly similar performance, with slightly declining similarity along the network hierarchy (i.e., low level representations are almost identical across networks, later stages differ more). We also compared our model against other network architectures (pre)trained on other datasets, finding that training the architecture studied here on the particular training set we used yields the closest correspondence to human judgments.

## Results

### Human viscosity ratings

We first sought to establish whether neural networks trained to estimate the physical viscosity parameter in computer simulations of liquids could predict human subjective viscosity judgments in such movies. To do this, we first measured human performance in a viscosity rating task, to establish the perceptual judgments against which the neural networks could be compared.

Sixteen observers each rated the viscosity of 800 movies of liquids, spanning sixteen viscosity steps across the ten scene classes. Within each scene class, five variations were simulated with different random parameters such as emitter velocity, geometry size or varying illumination conditions (see **Methods**). Viscosity ratings were provided via a response slider below the

stimulus, allowing the observer to report how runny or thick each liquid appeared. During training, observers were shown four example trials that included the maximum and minimum viscosities, to help them anchor their ratings.

**Fig 2** shows the results of the human observers (blue lines). Throughout, reported values are the means across the five variations of each scene. Some scenes (e.g., scene 1) yielded significantly better performance than others (e.g., scene 4 and scene 6). Overall, physical viscosity explained 68% of the variance in human ratings ($R^2 = 0.68$, $F(1,158) = 337$, $p < .001$). Previous studies have shown that humans can achieve much higher—indeed, almost perfect—accuracy [13–16]. However, compared to these studies, we had a different goal, in that we wanted the model to predict both errors and successes of the observers. Therefore, we used less detailed simulations and $64 \times 64$ images, to make viscosity perception more challenging in order to yield a higher proportion of perceptual errors. This also led to high inter-subject variance, with an average of 1.99 RMSE from each individual observer with the mean observer, an error size of 12% of the viscosity scale (**Fig 3A**). Yet, the overall pattern of responses across scenes spans both very good viscosity perception (e.g., Scene 5) and rather poor viscosity perception (e.g., Scene 6)

## Network predictions

Having established human performance across a range of conditions, we next trained neural networks (see **Methods: Network architecture**) to estimate viscosity in a training set of movies that did not include those shown to the participants. Our goal was to test whether such training would lead to internal representations that mimicked the pattern of successes and failures in the human judgments.

The predictions of one neural network is shown in **Fig 2A** (red lines). Overall the model has roughly the same performance as human observers in explaining the physical viscosity

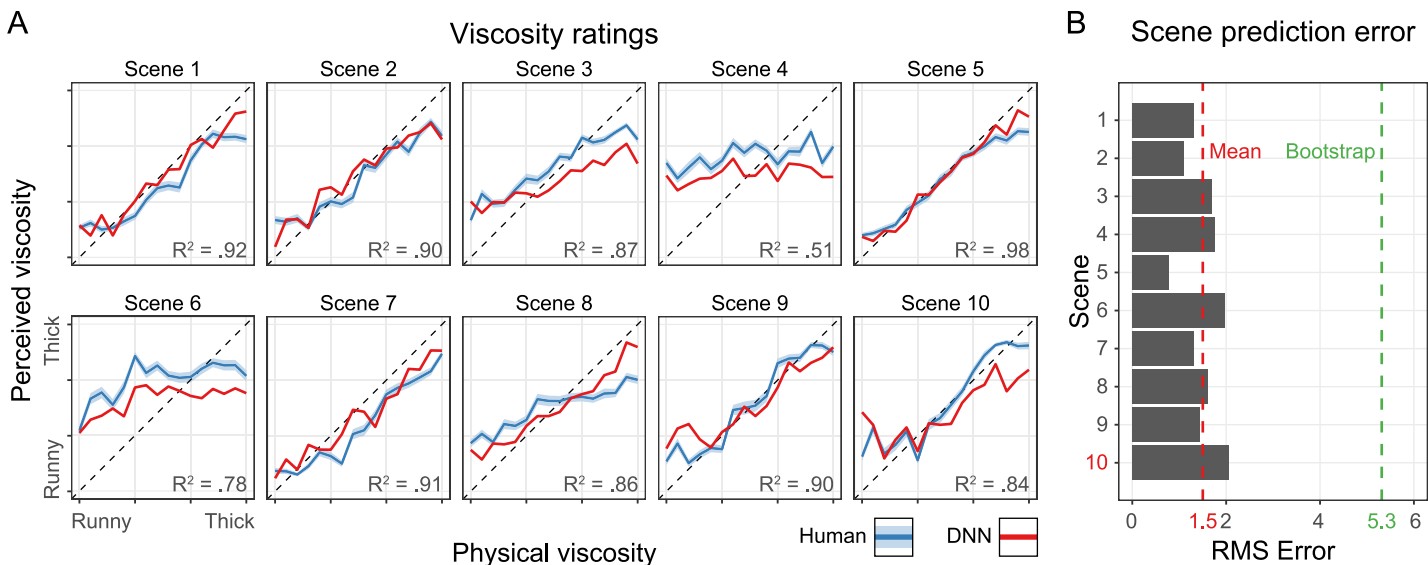

**Fig 2.** (A) Viscosity ratings for the 10 different scenes. The x-axis shows the physical viscosity steps (1–16). The y-axis shows the perceived/predicted viscosity averaged across the five variations. The error ribbons show the standard error of the mean (SEM). Blue lines are human viscosity ratings and red lines are DNN viscosity predictions. The dotted diagonal shows the physical truth. The DNN was not trained on any stimuli predicted here, and scene 10 (red) was completely left out of the training set in order to test generalization to other scenes. (B) The x-axis shows the Root Mean Square Error for each of the 10 scenes on the y-axis. This is the error between human observations and the network predictions. The red dotted line shows the mean error across scenes and the green dotted line the error of 1000 randomly drawn observations.

($R^2 = 0.73$, $F(1,158) = 437$, $p < .001$). Importantly, the network is very good at predicting the differences in viscosity perception across scenes. For example, like humans the network performs well with Scene 5 and poorly with Scene 4. Thus, the model correctly predicts both successes and failures of human perception. Indeed, the RMSE between the network's predictions and mean human judgments is only 1.50 viscosity units (on our 16-point scale), i.e., 9.39% of the total viscosity scale (**Fig 2B**).

Importantly, as a key test of generalization, when we trained the networks, we excluded the experimental stimulus set and all movies from Scene 10. Testing the generalization performance to these stimuli, which were never presented during training, reveals that the network does perform somewhat worse at predicting human perception for Scene 10 than for the other scenes, with a 36% larger error than the mean across scenes (RMSE = 2.05, i.e. 12.79%). Nevertheless, even for this set of stimuli—on which the network was never trained—the prediction error was still within the range of individual differences between observers.

To get a better sense of variability between networks we trained 100 instances of the same network, where only the random initialization and the randomized order of the training stimuli were different. The representative neural network we report throughout the paper is the network that best predicts the perceived viscosity in terms of error (network 78 of 100). However, overall, the different instances of the network yielded quite similar performance (**Fig 3**). We discuss the differences between networks in more detail in **Network Differences**.

Comparing individual observers, or the predictions of the representative network, with the human mean (**Fig 3**) reveals that the network performs better than all but one of the individual observers at predicting mean human judgments across observers in terms of error. The different instances of the network are closely clustered together and perform very similarly (Mean RMSE = 1.70, i.e. 10.63%, SD = 0.10). The representative network correlates highly with the human mean as well ($r_p$ (158) = .88, $p < .001$).

For comparison, we generated a bootstrap prediction based on 1000 random samples of ratings (RMSE = 3.75, i.e. 23.4%, $r_p$ (158) = .00, $p = .50$). In **Fig 3A** all data points are in the bottom half of the plot meaning that the error is larger to the physical truth than to the perceived

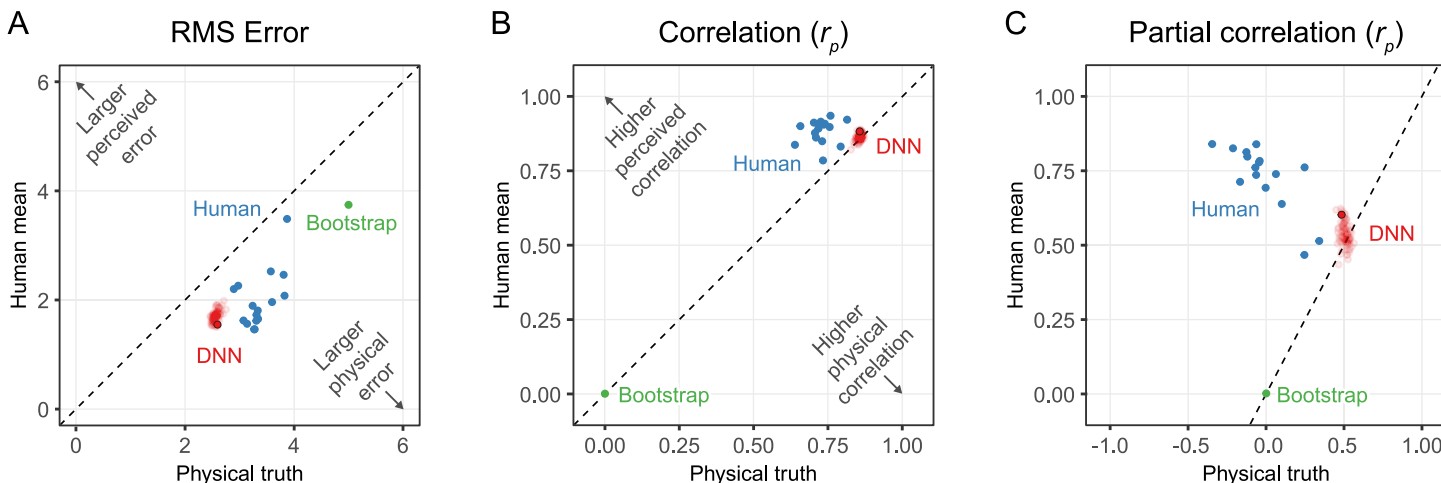

**Fig 3.** (A) Root Mean Square error of individual observers (blue), individually trained DNN networks where the final network has a black outline (red), and the green dot shows a bootstrapped estimate of random performance based on 1000 random draws. If data points are in the bottom half of the plot the error is larger for the physical truth than the human mean or perceived viscosity. (B) Same type of plot showing the Pearson correlation instead of RMSE. Partial correlations are performed with the human mean where the physical truth was the controlling variable. If data points are in the bottom half of the plot the correlation is larger with the physical truth than the human mean or perceived viscosity. (C) Same plot as B only with partial correlations where for the human mean the physical truth is a control variable and for the physical truth the human mean is a control variable, showing the independent correlations.

viscosity. This demonstrates that the network predictions are more similar to perceived viscosity than the physical viscosity. **Fig 3B** plots correlation instead of error, which reveals that networks are centred approximately evenly between the perceived and physical viscosity, similar to the individual observers. Importantly, however, physical and perceived viscosity are of course highly correlated. As a purer test of the extent to which the network predicts human perception, **Fig 3C** shows the partial correlations of these factors, where for the human mean the physical truth is the control variable and for the physical truth the human mean is the control variable. Here, we see that both individual observers and—to some lesser degree—the networks correlate with each other disregarding the variance explained by the physical truth. In particular, it is interesting to note that individual human observers hardly correlate with the physical viscosity, once the partial correlation with the human mean is factored out, while the networks do capture a component of the ground truth, independently of the human ratings. This is unsurprising as the networks were trained on ground truth, rather than human ratings. Indeed, the fact that the networks correlate so well with human judgments despite not being trained on them is somewhat surprising.

For these stimuli, viscosity estimation is challenging, as demonstrated by the low overall performance and large inter-observer variance compared to previous studies. Despite this, the neural networks seem to latch onto spatial and temporal image information that captures some core characteristics of the human judgments. Interestingly, further training actually reduces the network's ability to predict human perceived viscosity (**Fig 4**). Around epoch 30 is a pivotal moment after which overfitting starts to increase (i.e. blue curve separates from the green curve). There is still some improvement in physical viscosity estimation for the validation set, but the perceived viscosity prediction worsens from this point onward. Also interesting is the difference between the physical and perceived validation performance early in training, when the perceived validation performance improves more quickly than the physical validation.

Together, these findings show that we have developed an image computable model that predicts human perception in a challenging material perception task. In particular, we find that one approach to developing such a model is to train the neural network to estimate ground truth physical viscosity with tens of thousands of movies, while optimizing the network's hyperparameters via a Bayesian optimization to minimize the error in predicting the perceived

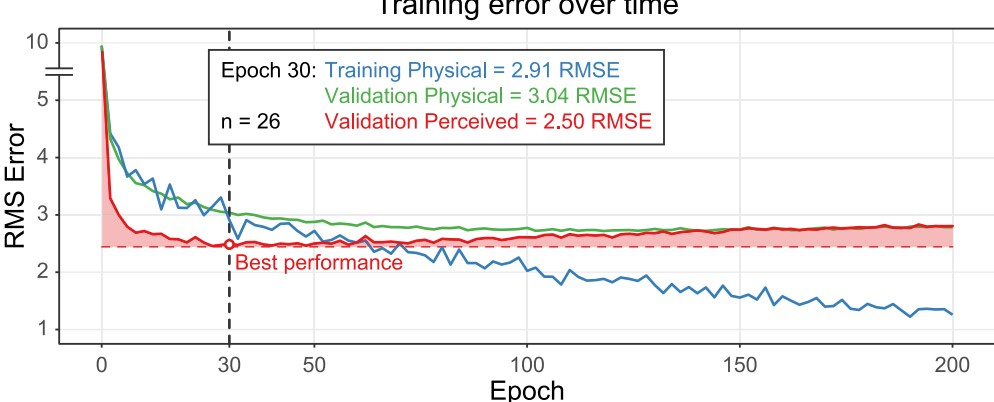

**Fig 4.** Mean training and validation error (y-axis) as training time increases (x-axis) across 26 individually trained networks. The 100 networks used in this study are trained for only 30 epochs since the perceived viscosity prediction error increases as training continues. The Validation Perceived shows larger errors than in the previous figures and results because here we do not average across variations for a clean comparison with the training results.

viscosity of the 800 experiment stimuli. Furthermore, we found that by training for a relatively short period of 30 epochs, training of the network is stopped at the optimal point for predicting human perception, while further training decreases performance. This partially overcomes the challenge of having sufficient labelled data to train directly on human judgments, and allows us to test the role of specific learning objectives and training sets in human performance. For further details see **Methods: Network architecture**.

## Neural activity

Having established that the networks provide a reasonably good model of human perceptual judgments, we next sought to investigate their inner workings. Specifically, to gain a better understanding of the computations performed by the networks, we carried out Representational Similarity Analysis for unit-level and layer-level activations, and Centred Kernel Alignment to compare activations between networks (RSA [18,19], and CKA [20]).

The neural activations are gathered at each ReLU layer in the network (see **Methods: Network Architecture**). In order to interpret the response patterns, we compare these neural activations with a set of *predictors*. These include 'hand-engineered' image metrics (describing colour properties, optical flow, spatial structure, etc.) as well higher-level predictors such as the physical viscosity or the scene class. To compare the responses of the network with each predictor, we used the experimental set of 800 stimuli.

**Unit activations.** To gain a detailed view of the network's responses—analogous to single-unit electrophysiology—we performed RSA at the level of individual units, mapping out how each unit in the network represents relationships between all 800 experimental stimuli and comparing these to image-based and high-level predictors (**Fig 5A**). Specifically, for each of the 800 stimuli, we gather single-unit neural activation patterns from the network; image feature values computed from each movie; and high-level features associated with each stimulus (e.g., perceived viscosity, scene label; **Fig 5B**). For each of these quantities, the differences between each of the 800 stimuli and all others are then computed and stored in a Representational Dissimilarity Matrix (RDM; **Fig 5C**). We then measure how well the RDM for each image feature correlates with the RDM derived from a particular unit in the network. In **Methods: Image Metrics** we give a full overview of the eighteen predictors we applied. We categorize our predictors in five classes, (1) motion, (2) spatial structures, (3) lightness and colour, (4) multi-feature models and (5) high-level predictors. Example predictors include optical flow gradients, local contrast, colour saturation, and GIST descriptors [21]. **S1 Fig** shows the correlations between the predictors themselves. Excluding the high-level predictors (viscosity and scene) we find that a RDM regression model of our image metrics only predict 2% of the perceived viscosity similarities ($R^2 = 0.02$, $F(1,13) = 376$, $p < .001$). Thus, such features are not sufficient on their own to account for human perceptual judgments. This reinforces the fact that viscosity perception is a nontrivial visual inference that cannot be achieved solely through straightforward image cues. It is nevertheless interesting to ask to what the network represents and builds on such features in order to estimate viscosity and emulate human judgments.

For each unit in the convolutional layers, we have a location in the eighteen-dimensional predictor space. **Fig 5D** shows a subset of the eighteen predictors for four example units, and their correlation between the RDMs of the predictors and the activation RDM of one unit. To get a clearer impression of the unit-specific function we visualized the stimuli that minimally and maximally activate the unit (**Fig 5E** and **S2 Video**).

Since we have the location of each neural unit in our 18-D predictor space we can visualize the similarities between the different units in 2D, using tSNE (**Fig 6**). To classify units, we applied a clustering analysis to the locations in the full 18-D space. Here we applied the

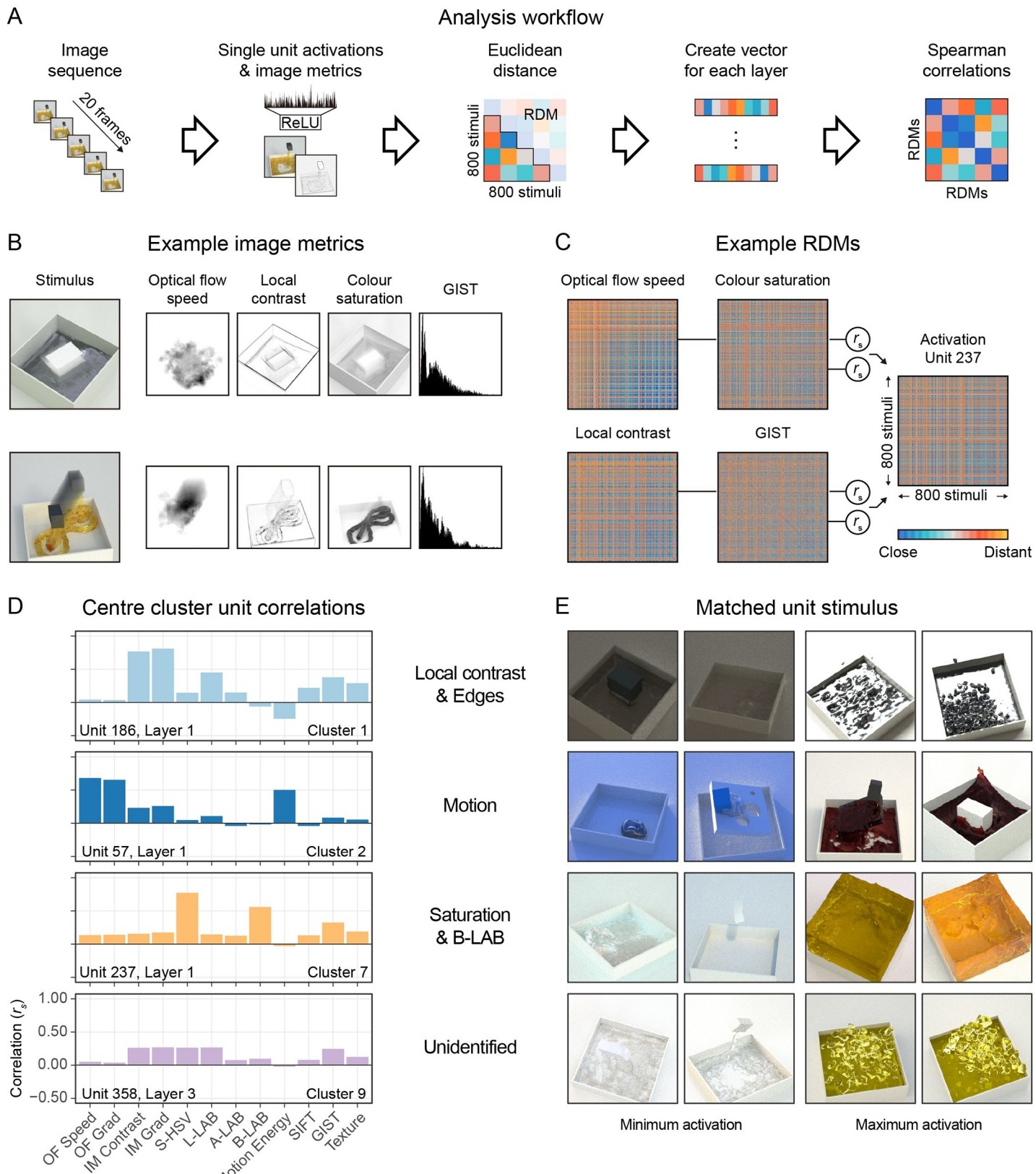

**Fig 5.** (A) RSA workflow for unit-level analysis. (B) Two stimuli with examples of the resulting image metric output. The ghosting effect shows motion over time. The multi-feature metrics such as Motion Energy and GIST lose the spatial structures. (C) Example RDMs of the same image metrics as B. Each row/column represents a stimulus, and colours indicate the distances between each pair of stimuli in terms of the corresponding image metric. Each RDM is correlated with the activation RDM of a single unit, in this case unit 237. (D) A selection of the RSA correlations for the units closest to the centres of four of the clusters shown in **Fig 6**. (E) The two stimuli of the entire dataset that created the minimum and maximum activation responses for the units of D.

Louvain clustering algorithm [22] which uses nearest neighbour weights to form communities, which project more intuitively onto the tSNE space than other clustering algorithms. The number of clusters was defined by the k-nearest neighbours algorithm, where the number of neighbours (k) in this case was 20, the square root of 420 units (256 in layer 1, 64 in layer 2, and 100 in layer 3). To interpret the functions of the different clusters, we manually assigned labels to them based on the mean (**S2 Fig**) and cluster centre correlations for the eighteen predictors. This analysis reveals how different units extract different kinds of information from the movie sequences. Not all clusters are equally interpretable, but units in Cluster 3, for example—most of which are found in Layer 1—seem to be strongly affected by image contrast, whereas units in Cluster 2 respond predominantly to the magnitude and type of image motion.

There is one cluster (9 in **Fig 6**) which contains many Layer 3 units and which is harder to identify. Cluster nine correlates poorly with all of the predictors (max $r_s$ = 0.17, min $r_s$ = -0.17),

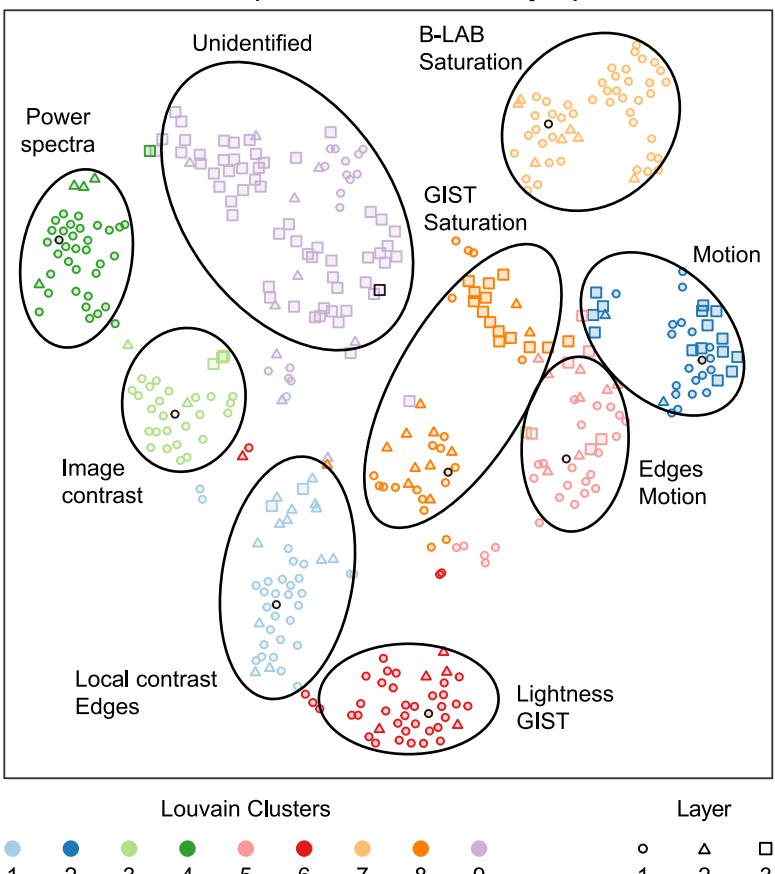

**Fig 6. tSNE plot showing all 420 units of the three convolution layers in eighteen-dimensional predictor space.** The units are colour coded showing nine Louvain clusters. We manually assigned labels to the clusters, that correlate well with specific image features or predictors. The centre of each cluster, defined by the largest mean weight with the units in the cluster, has a black outline.

including the high-level predictors (scene and viscosity), despite including many Layer 3 units. This suggests that there are other potentially important predictors, beyond those that we tested. To test the importance of cluster nine for the final viscosity prediction, we destroyed the 83 units it contains (artificial lesion). Destroying this cluster affects the prediction error by 3.43 standard deviations more than destroying 83 randomly selected units (n = 1000). Clearly cluster nine is crucial for good viscosity prediction and none of our predictors align with the functions it performs, including viscosity itself. This suggests that deeper in both artificial and biological visual processing streams lie units whose response characteristics are crucial for high-level visual tasks, but which may be difficult to describe in terms of conventional analyses or hypotheses about function.

**Unit visualizations through activation maximization.** As an alternative approach to gaining insights into the factors that drive single unit activity, we applied activation maximization to create visualizations of each unit's response function (**Fig 7**, **S3 Video**). The parallel pathways of the slow-fusion architecture allow temporally-specific features to be captured per pathway. This freedom regarding how temporal and spatial information is encoded, together with small kernel sizes, yields visualizations that tend to be abstract and difficult to interpret, compared to those that emerge in networks for classifying objects in static images [23–25]. Layer 1 and 2 have different temporal lengths with partial access to the full image sequence (i.e., L1 = 8 frames, L2 = 12 frames, L3 and L4 = full sequence of 20 frames).

Based on visual inspection, we find that the first layer mostly contains simple motion-related features of different temporal frequencies and orientations. Colour plays some role and varying degrees of lightness are also encoded. Layer 2 features seem to encode a range of textures with temporal and colour variations. These included both pulsing and flowing spatiotemporal textures with diverse directions. In layer 3, features consist of strongly contrasting textures in different spatial and temporal positions. Yet the responses become increasingly abstract and it is hard to imagine that such units are truly predictive of viscosity, suggesting that representations are highly distributed (i.e., rely on population activities across many units, rather than 'grandmother cells' for specific viscosities or flow patterns).

The visualisations from fully connected layer 4 mostly depict noisy patches with temporally recurring colour patterns that are synchronized across units. This synchronicity also occurs with varying seed images, suggesting that these colour sensitivities are similarly encoded across units of layer 4. This raises the question of whether temporal colour sequences might be an important cue for the network's function, even though within a given stimulus, colour remains

## Activation maximization

Conv1 - 256 channels    Conv2 - 64 channels    Conv3 - 100 channels    FC4 - 100 channels of 4096

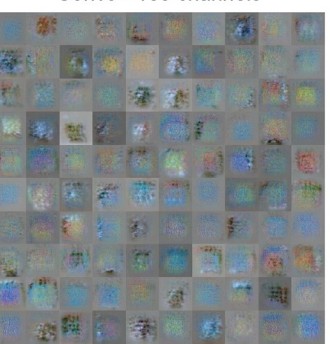
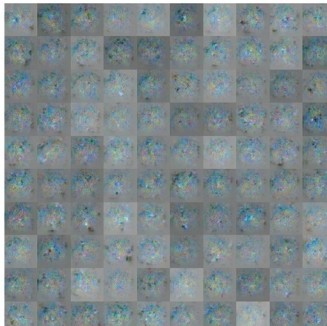

**Fig 7. Static snapshots of activation maximization results for each layer.** Layer fully connected 4 (FC) has 4096 units and we randomly picked 100 for this figure. We recommend **S3 Video** for visualizations of the temporal effects.

broadly constant, and in humans viscosity perception is largely independent of colour [15]. However, we find that the prediction error of the network increases by only 7% when we use grayscale stimuli. This suggests that the colour provides only limited information for viscosity estimation. Thus, the synchronized temporal fluctuations in colour sensitivity across units in layer 4 remain difficult to explain.

Despite some abstract representations in the deeper layers, earlier layers do depict properties that align with the unit-level RSA findings (**Fig 6**). Another interesting observation for Layer 2 onwards was the frequent recurrence of a large, approximately central spatial window, (presumably) related to the positions where most liquid-object interactions occurred, averaged across the entire stimulus set.

**Layer activations.** Another level of analysis for characterising network function considers the pooled activity of all units within each layer of the network. We next performed such population-level analyses to address the following two questions: (1) how do representations change along the network's hierarchy, and (2) to what extent do the representations depend on the network training set and objective function or on the network capacity (number of units).

**Fig 8** shows the Spearman correlations for each metric with each ReLU layer in the network (see **Methods**: **Network Architecture**). Here, the activations of all units in each layer are combined to form a layer RDM. Each bar shows how much the RDMs between layer activation patterns correlates with the RDM for each predictor. If analyses at the unit level can be thought of as analogous to single-unit electrophysiology, analyses at the layer level are loosely analogous to LFPs or even fMRI data in that they represent the responses of entire populations of units that may have extremely diverse responses. Indeed, given the diversity of responses from different units within a layer, it is unlikely for an entire layer to correlate highly with a given predictor, even if the layer contains individual units that do respond more strongly to a given feature. Despite this, there are a number of notable trends. First, despite the obvious importance of motion cues for viscosity perception [13] simple optical flow-based features predict layer activations surprisingly poorly at all layer depths.

Second, there is a general tendency for lower-level features, such as saturation and local contrast to correlate more strongly with the first and second layer than with deeper layers. In contrast, high-level properties, like the viscosity, and more importantly the perceived viscosity, correlate better with deeper layers than earlier layers. This is in line with our general understanding of the visual system, whereby more complex concepts or features are represented at later stages of the visual processing.

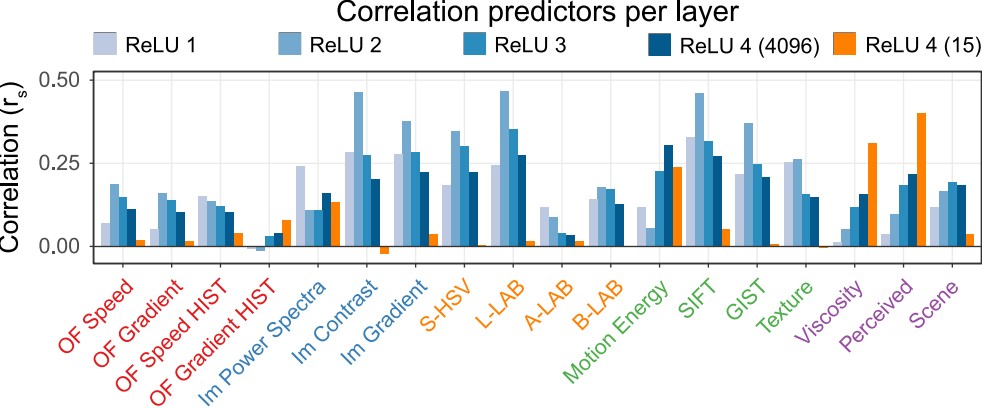

**Fig 8. The Spearman correlations for each layer and predictor.** We included two final layers, one with the original 4096 units and one where the last layer was retrained to contain only 15 units (see second half **Results: Layer activations**). Only the B-LAB predictor for layer ReLU 4–15 was not significant ($p > 0.05$).

Another observation is that despite this overall tendency, ReLU layer 4 (dark blue bars) still encodes many different features, including lower-level ones. Even though the network is trained to deliver viscosity estimates as output, many other properties, such as motion energy and luminance can be decoded from the penultimate layer of the network at least as well, if not better than the viscosity.

To verify these observations, we trained linear decoders at each layer to predict perceived viscosity and find trends in line with the correlation of perceived viscosity shown by RSA. Already by ReLU2, viscosity prediction error is reduced to RMSE of 2.57—a 12% larger error than the full network (RMSE = 2.30). ReLU3 encodes enough information to perform at the same level as ReLU4 in terms of prediction error. These findings further reinforce that viscosity estimation is a nontrivial visual inference. The features in the early stages of the network are insufficient to estimate viscosity to a level comparable to humans perception. It is only at later stages that such features of sufficient complexity emerge.

**Layer representations are strongly influenced by their capacity.** The finding that factors other than viscosity are encoded in ReLU 4 raises an intriguing question about the nature of the representation in the final stages of the network. To what extent is the representation determined by the demands of the objective function, and to what extent is it due to network architecture? One possibility is that representing other physical factors is actually necessary for succeeding at the objective. In other words, the network might need to explicitly represent the scene or some of its properties to disentangle viscosity correctly. On the other hand, the tendency to represent factors that are seemingly unimportant to the task might simply reflect 'excess representational capacity'. In other words, although not crucial for succeeding at the viscosity estimation objective, residual encoding of additional factors might come at no cost, given the high number of units available relative to the task.

To pit these two hypotheses against one another, we compressed the 4096-unit fully connected layer FC4 until the prediction performance started to decrease. To do this, we fixed the weights of all layers before FC4 and retrained different instances of FC4 while gradually decreasing the number of its units. We found that even with just 15 units in FC4 (i.e., a 273-fold decrease in the capacity of FC4) the prediction performance remained practically unchanged (perceived viscosity FC4-4096 = 2.30 RMSE vs. FC4-15 = 2.26 RMSE, **Fig 9**). The 15-unit layer is plotted in **Fig 8** in orange.

When we compare the correlations of the different features with the 4096-unit and 15-unit versions of FC4, we see some clear differences. Where 4096 units have the capacity to encode

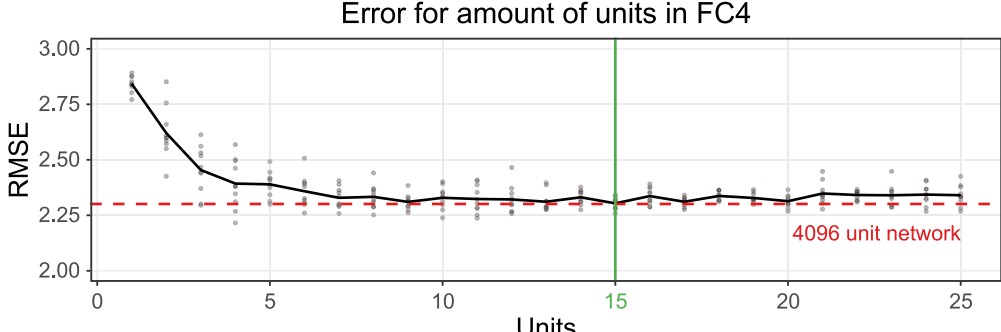

**Fig 9. Perceived viscosity prediction errors for networks that are retrained with a specific unit size in fully connected layer 4.** We retrained ten networks for each unit size, where all weights of the layers before FC4 were fixed. We finally selected the best performing 15-unit network as reference for our analysis. The 15-unit condition showed with our ten samples the lowest mean and low variation. Here we do not average across variations.

many features, 15 units lack the capacity for these richer descriptions and are forced to focus more on those stimulus characteristics that are strictly necessary to achieve accurate viscosity estimation. This is clearly demonstrated by the physical and perceived viscosity predictors, which are much more highly correlated with the 15-unit layer than the original 4096-unit layer, while the final output of the two networks is practically identical. This tendency is further exemplified by the scene predictor: the 4096-unit layer has sufficient capacity to encode more scene specific features, while scene can be decoded much less well from the 15-unit layer, which concentrates its representational resources on viscosity, as dictated by the objective function.

To further understand the differences between the 4096-unit and 15-unit layer we looked directly at the activation space of our 800 stimuli. **Fig 10A** shows the 2D tSNE plots of these activations. Each point represents a different stimulus, and the distance between points approximately indicates the similarity in network activation in layer ReLU4. With 4096 units, viscosity is encoded in a non-linear arrangement, with viscous and runny liquids represented in multiple groups. This means that the ReLU4 represents runny liquids as similar to one another, while thick liquids are different from runny ones, but also, are highly different from one another. In contrast, the 15-unit activation space shows a much more linear arrangement of viscosity. In the 4096-unit representation, there also seem to be separate clusters within each viscosity, whose origin becomes clearer when we colour code the same points by their scene class. For example, Scene 5 (pink), which was the best predicted scene, occupies its own corner in this space, with what appears to be its own local viscosity axis (**Fig 10C**). In contrast, the 15-unit activation space is much less sensitive to the scene.

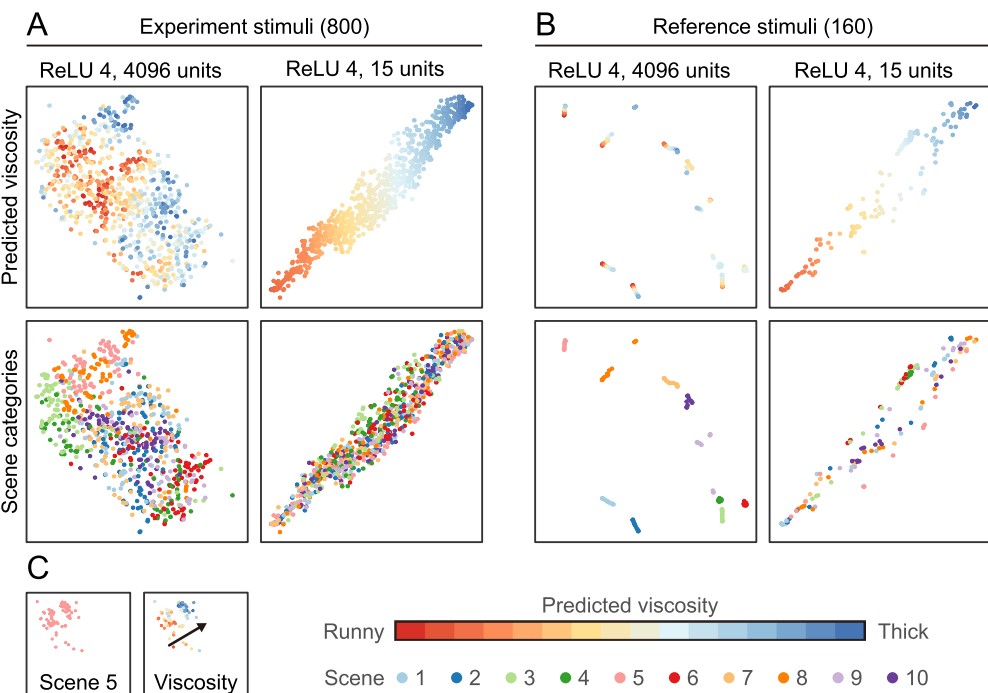

**Fig 10.** (A) The tSNE plots of the activation space of the two final layers, one with 4096 units and one with 15 units. This dimensionality reduction technique shows how the 800 stimuli are distributed in the final layers. Both predicted viscosity and the different scenes are plotted in the same space. (B) The same directly comparable tSNE space where instead of our 800 stimuli a reference set is used. This reference set only varies in viscosity, optical parameters are constant across scenes and viscosities. (C) Sub-selection and scaled down plot of the 4096-unit version of A. Here only scene 5 stimuli are fully visible.

To make these trends more clearly visible, we generated a new reference stimulus set of just 160 stimuli. All the random factors such as camera viewpoint, optical materials, and simulation settings such as varying emitter velocities were held constant. Only the sixteen different viscosity steps were simulated for each scene, yielding much more similar looking stimuli (**S3 Fig**). When we look at the placement of these more controlled stimuli within the same activation spaces (using the same tSNE space for direct comparison) we see clear scene-specific clusters in the 4096-unit condition (**Fig 10B**). Even more impressive is that each scene, especially the scenes that are predicted well, seem to have their own local viscosity scale. Generalization is also visible with Scene 10—which the network was never trained on—yet still has its own clearly defined cluster (dark purple), although viscosity is less linearly arranged than for some other scenes. In comparison, the 15-unit representation again shows a nearly one-dimensional viscosity trend.

It is important to emphasize again that the prediction performance of the 15-unit and 4096-unit versions of the network are practically identical, while we demonstrate here that the internal representation can vary dramatically on the capacity (i.e., degrees of freedom), in layer FC4. This demonstrates that when networks have capacity that exceeds the bare minimum required for the task, they may encode (i.e., retain, or fail to exclude) aspects of the stimuli that are not strictly necessary for task performance.

To test this, we measured the ability of the network to classify the different scene classes by applying transfer learning on FC4-4096 and subsequent layers. As before, the layers before FC4 —including all convolutional layers—remained unchanged. **S4 Fig** shows that the retrained network achieves an 88.62% classification accuracy across the 10 classes (AUC = 0.993) when only the final layers were retrained for eight epochs. This is measured across the 800 experimental stimuli in which many parameters vary (e.g. camera viewpoint, illumination, liquid velocity). As expected, we find that the FC4-15 performs worse—although not terribly—for scene classification (77.25% accuracy, AUC = 0.974). These findings demonstrate the power of the features in the earlier layers which can be repurposed to perform different tasks.

**Network differences.** The final level of analysis we considered was at the level of entire networks. One key question about the functioning of the networks is the extent to which they converge on similar internal representations. How many different solutions are consistent with human perceptual judgments? To investigate this question, we trained the same network architecture 100 times with identical training sets but different initial weights and training sequences to compare the resulting representations in the networks. We find that at the end of training, all 100 networks performed very similarly to humans (both in terms of correlation and prediction error; **Fig 3**, red dots).

To compare network representations in greater detail, we next applied Centred Kernel Alignment (CKA, [20]), which has proven to be especially accurate for comparing neural activity between networks. Very similar to the Pearson correlations in RSA, CKA uses the dot product between examples and additionally applies 'cocktail blank normalization' (i.e., subtracting the overall mean across observations for each observation) [26,27]. We performed this comparison for each layer and find that the networks converge on very similar neural activity (**Fig 11**). Especially for the first layer the similarity is extremely high—only marginally below 1.0— meaning that regardless of random initialization and the random shuffling of the training batches, networks converge on similar representations. The deeper layers show gradually less similarity with a similarity of 0.90 for the last layer. To investigate the differences between networks further we also performed the unit-level RSA (**S5 Fig**) across the four most dissimilar networks. This indeed shows very similar structures of functionality on the unit-level as well.

**Other network designs.** All of the analyses presented so far have concentrated on one particular network design (slow-fusion). Yet it is interesting to ask whether other networks

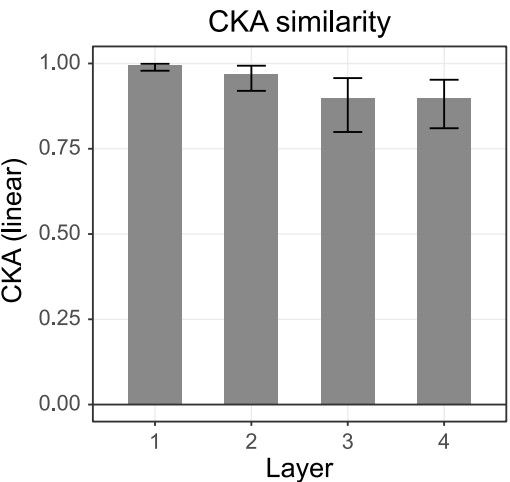

**Fig 11. The mean CKA similarity index across networks for the four ReLU layers.** Error bars show the 1st and 99th percentile.

might also perform well at predicting human viscosity perception. Research on video classification or regression networks is not as mature as, for example, networks for object classification in static images. The scarcity of labelled video datasets, computational costs associated with training, and input differences of both spatial and temporal resolution make it harder to compare networks directly. Nevertheless, to get an idea how our network measures against other spatiotemporal network architectures, we selected two action classification networks for comparison [28,29]. A number of well-documented datasets exist for action classification, making this the most developed video training task for which networks are currently being applied.

**Fig 12** shows the viscosity prediction errors of the S3D-G [28] and D3D network [29] which are evolved versions of the I3D model [30]. Both networks use inception modules with 3D convolutions that process spatial and temporal information separately. Applying action classification directly to our liquids dataset tends to yield plausible responses (of those available), such as 'making tea' or 'cleaning toilet' (i.e., actions involving liquids). We next sought to test how well the features learned by the action classification networks can be repurposed for a viscosity estimation task, and to what extent they predict human viscosity perception. To test this, we used transfer learning.

Specifically, as with our previous transfer learning analyses (see **Layer Activations**), we trained a linear decoder to predict the physical viscosity using the neural activity of the final layer before the prediction layers of these networks (i.e., mixed5c). The decoder contains 12288 weights and was trained for 30 epochs using gradient descent. We find that the action classification networks perform and generalize quite well, both in terms of estimating physical viscosity and in terms of predicting human perception. Nevertheless, the best performing D3D K400 network has a 10% larger error than our own network for the validation set with perceived viscosity labels. A similar trend with larger errors for the physical viscosity labels is observed. These findings demonstrate that action recognition networks do learn features that are somewhat useful for viscosity estimation as well, further reinforcing the notion that viscosity perception can draw on general-purpose cues and measurements. At the same time, training on diverse and naturalistic stimuli is not necessary for successfully predicting human perception. While our network has a single purpose design, and is trained exclusively on computer graphics imagery, in comparison with other networks, it is competitive at estimating physical viscosity and predicting human viscosity judgments.

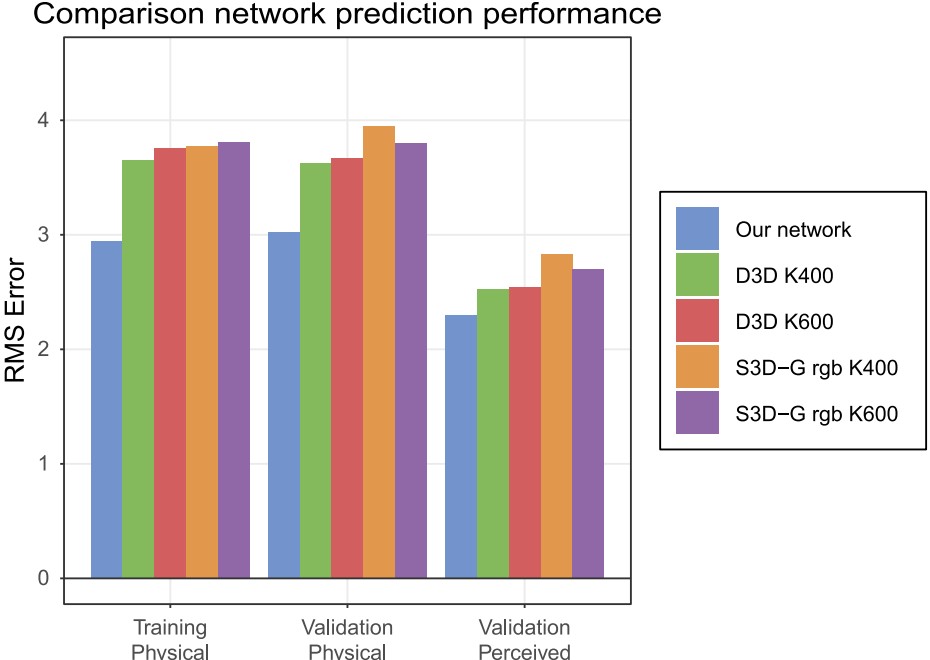

**Fig 12. Viscosity prediction error for two (D3D and S3D-G) video action video classification networks.** These networks were trained on two different datasets, the Kinetics 400 and Kinetics 600 datasets [31] where the numbers represent the number of classes in the dataset. For S3D-G we used the pathway that uses RGB images as input, this particular network has a pathway using optical flow input as well. As in **Fig 4**, we report training error using physical viscosity labels (average of last ten batches), and the validation error for both physical and perceived viscosity labels.

## Discussion

Visually estimating the viscosity of liquids is challenging, especially in short, low-resolution animations depicting liquids undergoing a wide range of different behaviours. Until now there was no image-computable model of human viscosity perception. The main contribution of our work is the demonstration that neural networks trained to estimate physical viscosity—but with hyper-parameters fine-tuned to match human performance—can quite accurately predict both the successes and scene-specific pattern of errors in human viscosity perception in our experiments. Thus, the relatively shallow feedforward networks tested here appear to capture some important aspects of the way observers perform the task. We reasoned that such a model is a useful experimental platform for investigating potential mechanisms of human perception. Accordingly, we then probed the network in a series of analyses to reveal which features were responsible for its performance.

Our networks were trained for only 30 epochs, which is a relatively short time. We found that after epoch 30 the perceived viscosity predictions worsened and the networks increasingly started to overfit (**Fig 4**). This is further demonstrated by the increasing difference between training error with physical viscosity labels and validation error with physical viscosity labels after epoch 30. It is interesting to speculate about the causes and implications of this finding of a U-shaped approximation to human performance as a function of training. For example, it could be an artefact of the training set. While humans learn to see from diverse and naturalistic stimuli, the models considered here were trained exclusively on computer simulations of liquids. It could be that with more varied, larger or natural training data, the approximation to human performance would continue to improve with further training on the physical estimation task (i.e., no U-shaped approximation to human performance would be observed).

However, another possibility is that the cues used by human observers are those that network also tends to learn first. It could be these cues are the most discriminable or most robust [32–34] cues of the dataset. As training continues, the network continues to improve at the physical viscosity estimation objective, possibly by learning more subtle cues specific to the dataset which the human visual system cannot discern at all or is less sensitive to. There is indeed some support for this speculation. Other studies that look into the early phase of neural network learning find critical learning periods similar to biological networks [35,36]. Evidence suggests that neural connectivity is broadly settled in a memory formation phase early in training, after which neuroplasticity decreases and only much smaller changes occur by reorganizing or forgetting less predictive weights [37,38]. This makes the early phase ($<$ Epoch 10) an especially critical period where dominant qualities of the dataset are encoded. In our case this period is defined by an especially large drop in perceived viscosity error. This is in line with our speculation that the most discernible cues which are encoded in early training align particularly well with the perceived viscosity cues used by humans.

To probe the networks' inner workings, we used RSA to compare neural activations with a range of image metrics that are easier to interpret and understand than the raw features of the trained network. We found nine clusters of units, each performing different classes of tasks (e.g., colour detection, motion detection, edge detection). Importantly, one cluster—consisting mainly of units in the deeper layers of the network—did not correlate with any of our predictors, yet was particularly crucial for the functioning of the network. This finding is arguably the second main contribution of our study. It suggests that there are one or more nontrivial features—as yet unidentified—which play a key role in viscosity estimation. We speculate that these features might be related to 'mid-level' shape and motion features (e.g. spread, splash, piling up), which we have found in previous psychophysical studies to explain a substantial proportion of human performance [13–16]. A major challenge for future research is to develop image-computable measurements that isolate and identify such mid-level features so that we can test the extent to which they are represented by the network. An alternative approach would be to ask human observers, or a separate network trained on human-labelled data, to estimate the mid-level features from the same movies as are used for probing the viscosity estimation network.

Other techniques such as deep dreaming [39] and activation maximization [24] provide means to coerce units in the network to produce images that elicit strong or weak responses from them, an approach that has yielded some insights into the features driving object classification networks (e.g. nostrils of dogs, seams in baseballs). However, in many cases—including in our analyses (**Fig 7, S3 Video**)—the crucial features may be spatiotemporally distributed and abstract (e.g. specific motion textures, 'clumpiness') and the images regurgitated from the network quickly become difficult to interpret. However, using such networks as a generative tool to create stimuli that are adversarial for human perception (i.e., creating novel illusions) might yield more useful insights.

Mapping and visualising relationships between stimuli in the network's pattern of activations is another way to probe underlying representation in the network. Using this approach, we found some clear and interpretable structures in the fully-connected layers at the final stages of the network. In particular, in the full 4096-dimensional representation, we found stimuli to be clustered by the scene, each with clearly-defined but distinct local viscosity axes. Scene 10—which was completely left out of the training set in order to test generalization performance—also project to a clearly identifiable location within activity space, with its own local viscosity axis.

The finding that factors other than viscosity were significantly represented in the final stages, despite not being directly relevant to the training objective, prompted us to investigate

the effects of network capacity on the internal representations. We hypothesised that this was a consequence of 'excess' capacity in the network beyond the bare minimum required for the viscosity regression objective. Would squeezing down the capacity (number of units) eliminate this effect without changing overall performance, or were the representation of other scene factors crucial to the success of the network? We found that the nature of the networks' final representations can vary widely depending on the number of units, practically independently of task performance. Surprisingly, the prediction performance is practically the same for FC layers with just 15 units, 4096 units, or anything in between. Yet, with only 15 units the FC layer primarily encodes viscosity to the exclusion of other aspects of the stimulus such as colour, lighting or scene. In contrast, increasing the capacity results in more local and nonlinear representations of viscosity within the feature space, along with representations of additional stimulus characteristics that are not strictly necessary for viscosity estimation.

Since representing these additional characteristics neither helps nor hinders task performance, the sensitivity to seemingly redundant features is likely a by-product of having excess representational capacity in the network. This suggests caution is necessary in drawing conclusions about biological representations from neural network models, as quite different representations at a given stage of the network can yield near identical decisions by the system as a whole. These findings also demonstrate that early layer features not only contain powerful processing capabilities for the given task, but as a by-product are descriptive enough to provide a foundation for inferring a rich variety of other high-level scene factors. This is further corroborated by the good transfer learning performance in which only the final layers were retrained for a different task. It supports the idea that earlier layers have the tendency to converge on task-invariant image representations: a basic toolbox of filters that dissect visual information for further processing in a wide range of visual tasks and are similar to the earlier processing stages in our visual cortex [40–45]. While this requires further research, we speculate that this is not a universal characteristic of DNNs, but an emergent property of complex models capable of learning a challenging task on an ecologically plausible training set.

We also investigated multiple instances of networks performing the same task. This analysis revealed strong correlations between the different instances of the network. The divergence between networks steadily increased for the deeper layers, similar to findings of Kornblith et. al. [20]. We did not find distinct clusters of networks that encode stimuli in an equally similar or dissimilar way.

Artificial neural networks have changed how we model visual and neural processes. Feedforward neural networks—despite lacking top-down and lateral processing [46,47]—have proven to be an insightful tool for vision science [48–56] and currently provide the most successful computer vision models as well. Many processing aspects of the human visual system are not properly simulated by feedforward designs and the sensory input methods and learning regimen leave much to be desired as well [57–62]. However, especially in controllable single-purpose scenarios, feedforward designs allow us to confirm or alter our hypotheses originating from psychophysical studies in insightful ways. For example, studies that compare brain imaging data with artificial networks show remarkable similarities, especially in early visual processing [63–66], suggesting some legitimacy in feedforward approaches.

In conclusion, we have developed the first image-computable model of human viscosity perception, which predicts average perceptual judgments as well as individual observers do. Our analyses reveal that the model uses a variety spatiotemporal features to encode the stimuli in a high-dimensional space, in such a way that subsequent stages can 'read out' the viscosity of the stimuli. The nature of the internal representation is complex, distributed and varies depending on the capacity of the network. This requires further research and leads to the intriguing speculation that cortical visual representations might be as much the result of the

*number of cells* in the different brain regions, as the specific task(s) the visual system learns. While some elements of the model can be easily interpreted in terms of familiar features—like contrast or motion energy—we also found that some of the components that are most important for successful prediction of perception are not easily expressed in terms of such familiar 'cues'. Indeed, like other studies of representations in neural networks, our research program hints that it might be time for the field to move away from the concept of mappings between a small number of clearly interpretable 'visual cues' and physical properties of the outside world. Instead, visual perception might be better formulated as a process of progressive 'disentanglement' in high-dimensional feature spaces [67–69], in which individual features are less important than the combined effect of the sequence of non-linear transformations.

## Methods

### Stimuli

To generate a large training set, we used computer graphics simulations of liquids interacting with ten different classes of scene. Each scene class exhibited specific liquid interactions (e.g. dipping, rain, stirring, spraying over various geometries), with variable parameters (e.g., lighting, trajectories of moving objects). Each scene was simulated with sixteen different viscosities values from a logarithmically spaced scale from 0.001 Pa·s to 10 Pa·s (roughly equivalent to a range from water to molasses). For training labels, we referred to these on a linear scale from one to sixteen. Each scene and viscosity were simulated several times to create the large quantity of movies necessary for training a DNN. Parameters such as liquid emitter velocity, emitter direction, initial liquid volumes and scene geometries that interact with the liquid were randomized. This process was repeated 125 times for each scene, and for these 125 variations, five different render variations were made, changing illumination maps, optical material properties of both liquid and scene geometries, and camera position. Twenty sequential frames were rendered providing moving stimuli of a 0.67 second duration (30 frames per second). This resulted in a training set of 20.000 unique simulations and 2 million images (10 scenes × 16 viscosities × 125 scene variations × 5 optical variations × 20 frames). A subset of this data was used for experiments with human observers, 800 in total (10 scenes × 16 viscosities × 5 scene variations).

**Simulation.**   The stimuli were generated using RealFlow 2015 (V. 9.1.2.0193; NextLimit Technologies, Madrid, Spain). Viscosity values were selected from a logarithmically spaced scale of 16-steps between 0.001 Pa·s and 10 Pa·s. The "Hybrido" particle solver was used, which simulates the dynamic viscosity of the liquids in real physical units (Pa·s). Hybrido is a FLIP (Fluid-Implicit Particle) solver using a hybrid grid and particle technique to compute a numerical solution to the Navier-Stokes equations describing viscous fluid flow. A meshing algorithm uses the particles to calculate the fluid boundary and creates a mesh. The density of the liquids was held constant at one kilogram per litre and gravity was the only simulated external force. The simulated animations had a total duration of four seconds (120 frames at 30 fps). Only the last twenty frames were used for the final stimuli. Each scene had specific parameters that were randomly assigned for each simulation. The random values were drawn from predefined ranges to limit the occurrence of artefacts. For example, in some scenes the liquid emitter changed position during simulation, where the initial position, size, rotation, and trajectory of the emitter were randomly assigned. The simulation space for each scene was one cubic meter. The white container in the scenes was placed on the simulation border making this container $1m^2$ large. The height of the container changed depending on the scene.

**Rendering.**   The render engine used to generate the final image frames was Maxwell (V. 3.0.1.3; NextLimit Technologies, Madrid, Spain). This render engine is built into Realflow

2015. The images were rendered at a $64 \times 64$ resolution where the sampling rate was kept lower than typically used to save time generating the 2 million images. Because of the lower sampling rate some noise was visible, yet visual inspection reveals this had negligible effect on perceived liquid properties. The illumination maps were randomly assigned from a set of 234 light probes, which were normalized and white balanced. The illumination maps came from diverse sources, some from scientific databases [70,71]. There were two categories of optical materials, solids (12, e.g. paints in various colours) and liquids (13, e.g. beer, chocolate milk, wine), which were randomly assigned to the different objects in a scene. At least one liquid object was present in every scene.

## Observers

Sixteen observers participated in the experiment (twelve female, four male). Mean observer age was 25.9 (SD = 3.6). All observers reported having normal or corrected-to-normal vision. All observers gave written consent prior to the experiment and were paid for participating. Experiments were conducted in accordance with the Declaration of Helsinki and prior approval was obtained from the local ethics committee of Giessen University. Six observers participated in an experiment with a static condition before performing the experiment with moving stimuli. With enough time, observers could participate in two experiments, both the static and the moving condition with stimuli of the same size. In this case the static condition was always performed first since these stimuli were less informative of the depicted liquid. Data from the static condition are not reported here.

## Procedure

The experiment reported here was performed with $64 \times 64$ pixels 30 fps movies. The experimental setup consisted of a Dell T3500 system running Matlab 2015a (v. 8.5.0.197613) and the Psychtoolbox library (v. 3.0.12) [72,73]. The stimuli were displayed on an Eizo ColorEdge CG277 27-inch monitor with a resolution of $2560 \times 1440$ and a gamma of 2.2. A training session was performed to get the observers acquainted with the task and interface. The training session consisted of four trials in which the maximum and minimum viscosity were included. The task was to rate the viscosity of the liquid in each stimulus, by adjusting a horizontal rating bar below the stimulus. The rating bar marker reacted to the x-position of the mouse. Once the marker on the rating bar was at the desired position, the observer confirmed the response by pressing 'space' on the keyboard, after which a new trial was loaded. In total 800 trials were tested, 10 scenes $\times$ 16 viscosities $\times$ 5 variations. There was no time limit for the trials.

## Network architecture

We applied a slow fusion model [17] implemented in Matlab 2017b (v. 9.3.0.713579) including the deep learning toolbox. In a slow fusion architecture there are parallel pathways that slowly fuse over time providing the higher layers with more global information in both spatial and temporal domains (**Fig 13**). Each pathway has a specific part of the image sequence as input. Between the pathways there is an overlap of input images; in our case, for the first convolutional layer the temporal extent T = 8 with stride 4 and for the second convolutional layer T = 12 and stride 4 of the original sequence. The third convolutional layer has access to the full input range of 20 frames. This is followed by a fully connected layer with 4096 units, a dropout layer with a dropout probability of 50%, and another fully connected layer with one unit for the regression output. The learning rate was set to $1.110510 \times 10^{-5}$, momentum to 0.43325, and L2 regularization to $4 \times 10^{-9}$.

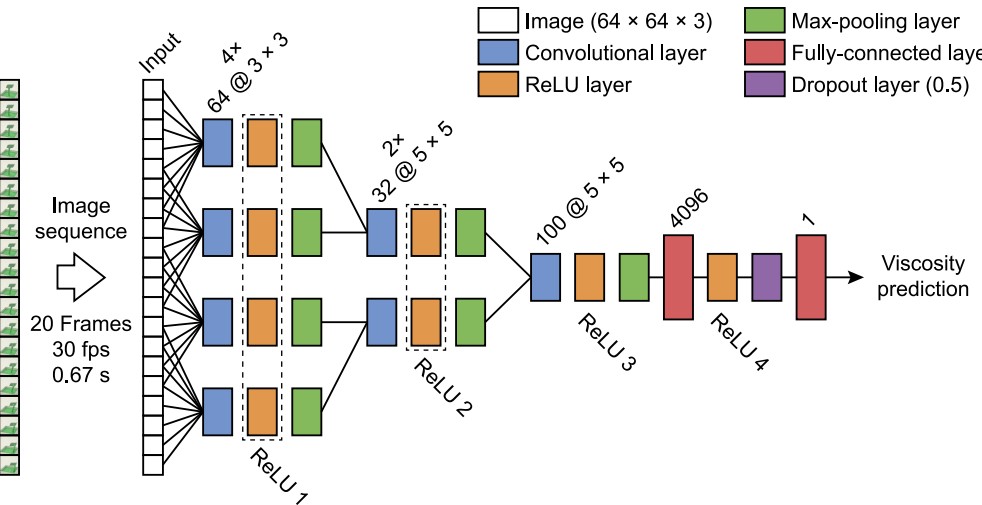

**Fig 13. The slow-fusion network architecture.** Input consists of a 20-frame animation of $64 \times 64 \times 3$ images. There are three sequential convolutional stages, although in practice multiple convolutional layers are placed in parallel at each stage. All neural activations reported here are measured at the ReLU layers of which the responses are combined for the parallel layers. The dropout layer randomly sets input elements to zero with a 50% probability during training.

The 800 stimuli of the experiment and scene 10 were excluded from the training set and were used for network validation. Bayesian optimization was used to determine the optimal settings for the hyper parameters; in this case the learning rate, L2 regularization, momentum, kernel sizes and filters for the three convolutional layers and the dropout probability of the dropout layer. The Bayesian optimization was fitted to the human labels of the 800 stimuli. This means that the hyper-parameters were set to achieve lowest error predicting human labels, not the physical viscosity labels on which the network was trained. The Bayesian optimization ran for 52 iterations of 25 epochs after which the most optimal parameter settings were provided. The optimal settings were close to the minima of the space we searched, the maxima produced networks that performed 47% worse in terms of perceived viscosity predictions.

## Representational similarity analysis (RSA)

We applied RSA on different scales of activation patterns. Here we provide specific details of choices we made during these analyses.

**Unit-level.** On the smallest scale we compare activation patters of specific units in one of the first three ReLU layers. For this analysis we concentrate only on units from the three convolutional layers. The 4096-unit fully connected layer was excluded from this analysis because there were units that did not get activated at all by any of our 800 experiment stimuli. For this analysis we used Euclidian distances instead for Pearson correlations since some units were not activated by one or more of our 800 stimuli. For the image metrics, which are described in detail in **Image Metrics**, we used the Euclidian distances as well. We calculate the Spearman correlations between the Euclidian based RDMs. These Spearman correlations are used to form an 18-D feature space that is shared by the 420 units in the network (**Fig 6**).

**Layer-level.** On the layer level activation patterns across an entire layer are used. Since an entire layer always has some activation for each of our 800 stimuli we switched from Euclidean distances to Pearson correlations to calculate the RDMs. Dissimilarity was calculated as 1-Pearson correlation. In this analysis, the fully connected layer was included. To make the activation data more comparable between convolutional layers and the fully connected layer,

we decided to measure the activations at the one common stage for both, in this case the ReLU layers. In this case for our image metrics we used the Pearson correlations as well. The only exceptions are the high-level metrics (e.g. perceived viscosity, scene) for which we used the Euclidian distances again since these are single value representations. The RDMs are then correlated using Spearman correlations resulting in **Fig 8**.

## Clustering

For the network similarity analysis, we applied a divisive hierarchical clustering algorithm (cluster v 2.0.9. library for R) since this is often used for distance matrices. Elbow, silhouette, and maximum likelihood estimation criterions showed no signs of significant clustering. In the case of the unit-level analysis we used the Louvain clustering algorithm [22], as it corresponded closely to the output of the tSNE analysis. This algorithm uses a predefined number of nearest neighbours to form communities (igraph v1.2.4 library for R). We applied the clustering in the full 18-D space and not the 2D tSNE space. The neighbours were identified using a k-nearest neighbours algorithm, where $k$, the number of neighbours was set to 20, the square root of 420 units.

## Activation maximization

We applied activation maximization to visualize features that maximally activate units of the network (**Fig 7**, **S3 Video**). A seed image is input into the network, and gradient ascent is used iteratively to modify the pixel values in the seed image in such a way as to maximise the activation of the given unit. The parallel pathways of the slow-fusion network with varying temporal dimensions and small kernel sizes made it particularly challenging to visualize interpretable features. There are various methods to improve the interpretability of the activation maximization results [24]. We achieved best results using the mean image of the entire dataset with added Gaussian white noise as the seed images for activation maximization. The noise was applied for each neural unit separately, varying the seed images. The activation maximization (Matlab function *deepDreamImage*) ran for 2000 iterations with one pyramid level. This analysis was performed using Matlab (2019b) and the Deep Learning Toolbox v13.0.

## Image metrics

For the RSA, we selected a wide range of image metrics that correlate to various extents with units and layers in the network. During pilot work, we also tested a larger range of image measurements but found that not all were relevant for the computations performed by the network or were too similar to other metrics. The final selection consists of eighteen metrics that describe both image and motion features.

**OF Speed:** The optical flow was calculated using the iterated pyramidal Lucas-Kanade method [74]. The flow vector length was used for the speed. Optical flow was calculated between consecutive frames. The rendering parameters led to spatiotemporal noise in some sequences to which the optical flow algorithm was sensitive. To counteract this, a mask was applied on the original image to evaluate optical flow only for those pixels within the perimeter of the box in the scene, where there was always liquid-based motion. The final stimulus representation has a $32 \times 32 \times 19$ size.

**OF Gradient:** The numerical gradient was calculated in both horizontal and vertical directions using the optical flow speed. The final stimulus representation has a $32 \times 32 \times 19$ size.

**OF Speed HIST:** The histogram of the optical flow speed using 44 bins. The optimal number of bins was estimated using the Freedman-Diaconis rule [75]. Since each stimulus often

results in an different optimal number of bins, we averaged the number of bins across all experimental stimuli. The final stimulus representation has a 44 × 19 size.

**OF Gradient HIST:** Similar to the optical flow speed histograms only using the gradient information (i.e., a measure of the distribution of accelerations in the stimulus). The final stimulus representation has a 44 × 19 size.

**Im Power Spectra:** The power spectrum calculated per frame of the sequence. The power spectrum is calculated using the grayscale stimulus and is rotationally averaged. The final stimulus representation has a 33 × 20 size.

**Im Contrast:** The local contrast was calculated using the range value (i.e. maximum–minimum value) of a 3-by-3 neighbourhood around the corresponding pixel of the grayscale input image. The final stimulus representation has a 64 × 64 × 20 size.

**Im Gradient:** The numerical gradient was calculated in both horizontal and vertical directions using a converted grayscale image. The final stimulus representation has a 64 × 64 × 20 size.

**S-HSV:** The saturation channel of the original stimulus converted to the HSV colour space. The final stimulus representation has a 64 × 64 × 20 size.

**L-LAB:** The lightness channel of the original stimulus converted to the CIE 1976 L*a*b* colour space. The final stimulus representation has a 64 × 64 × 20 size.

**A-LAB:** The A colour dimension of the original stimulus converted to the CIE 1976 L*a*b* colour space. The final stimulus representation has a 64 × 64 × 20 size.

**B-LAB:** The B colour dimension of the original stimulus converted to the CIE 1976 L*a*b* colour space. The final stimulus representation has a 64 × 64 × 20 size.

**Motion Energy:** The motion energy was calculated using the motion energy model suggested by [76,77]. In our case 6555 wavelets were used in the model. We used a Matlab implementation of the model written by Shinji Nishimoto [78]. The final stimulus representation has a 6555 × 20 size.

**SIFT:** Computer vision model that uses scale-invariant descriptors of an image for local feature detection and object recognition. A Matlab implementation written by Aditya Khosla was used to calculate the descriptors [79]. The final stimulus representation has a 29440 × 20 size.

**GIST:** A global vision model that describes orientations, colours and intensities on different spatial scales. A Matlab implementation written by Aditya Khosla was used to calculate the descriptors [79]. The final stimulus representation has a 512 × 20 size.

**Texture:** The texture descriptors of the Portilla and Simoncelli colour texture synthesis model [80]. The final stimulus representation has a 4455 × 20 size.

In the RSA, additional higher-level stimulus properties (i.e., not image computable, but derived from ground truth knowledge of the physical scene) were added as predictors as well.

**Viscosity:** The physically simulated viscosity value of the stimulus (on the 16-step linear scale, not the original logarithmic Pa·s scale).

**Perceived:** The perceived viscosity value of the stimulus (on the 16-step linear scale, derived from observer responses).

**Scene:** A binary map of stimuli that are from the same scene class (from 1–10).

## Supporting information

**S1 Video. Stimuli of moving liquids of the ten different scenes from our dataset.** The videos depict viscosities 1, 8, and 16 of our 16-step, runny to thick, viscosity range.
(MP4)

**S2 Video. The stimuli that minimally and maximally activate the centre units of each cluster.**
(MP4)

**S3 Video. Unit visualizations using activation maximization.** Visualization of Layer 1 (256 units), Layer 2 (64 units), Layer 3 (100 units), and Layer 4 (random 100 units out of 4096).
(MP4)

**S1 Fig. Correlation matrix of the different image metrics applied in our RSA analysis.** The Spearman's rank-order correlation ($r_s$) was used.
(EPS)

**S2 Fig. Mean cluster correlations of the RSA predictors.** On the x-axis the eighteen predictors and on the y-axis the Spearman's rank-order correlation ($r_s$).
(EPS)

**S3 Fig. Showing the stimuli from the reference stimuli set.** In this case all optical and physical parameters are held constant across stimuli except for the changes in viscosity making the images much more similar.
(TIF)

**S4 Fig. The scene classification probabilities for the scene transfer learning test, performed with our standard 800 stimuli.** X-axis show the classification matches and the y-axis show the test classes.
(EPS)

**S5 Fig. tSNE plots showing all 420 units of the three convolution layers in eighteen-dimensional predictor space.** The four most dissimilar networks from our standard network 78 (Fig 6) are shown. The units are colour coded showing the Louvain clusters. With the same workflow applied for each network the ideal amount of clusters can vary. We manually assigned labels to the clusters, that correlate well with specific image features or predictors. The circles are added for visual clarification. Across all networks we see similar functional clusters appearing in our eighteen-dimensional space. The function of layer three units are often hard to identify within this space.
(EPS)

## Acknowledgments

We thank Peter Battaglia for invaluable discussions and help getting started with deep learning and Next Limit for providing support during the stimuli generation process.

## Author Contributions

**Conceptualization:** Jan Jaap R. van Assen, Shin'ya Nishida, Roland W. Fleming.

**Data curation:** Jan Jaap R. van Assen.

**Formal analysis:** Jan Jaap R. van Assen.

**Funding acquisition:** Shin'ya Nishida, Roland W. Fleming.

**Investigation:** Jan Jaap R. van Assen, Shin'ya Nishida, Roland W. Fleming.

**Methodology:** Jan Jaap R. van Assen, Shin'ya Nishida, Roland W. Fleming.

**Project administration:** Shin'ya Nishida, Roland W. Fleming.

**Resources:** Jan Jaap R. van Assen, Shin'ya Nishida, Roland W. Fleming.

**Software:** Jan Jaap R. van Assen.

**Supervision:** Shin'ya Nishida, Roland W. Fleming.

**Validation:** Jan Jaap R. van Assen, Shin'ya Nishida, Roland W. Fleming.

**Visualization:** Jan Jaap R. van Assen.

**Writing – original draft:** Jan Jaap R. van Assen, Shin'ya Nishida, Roland W. Fleming.

**Writing – review & editing:** Jan Jaap R. van Assen, Shin'ya Nishida, Roland W. Fleming.

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
