## [Decision Letter · Decision Letter 0]

20 Jan 2020

Dear Dr. van Assen,

Thank you very much for submitting your manuscript "Visual perception of liquids: insights from deep neural networks" for consideration at PLOS Computational Biology.

As with all papers reviewed by the journal, your manuscript was reviewed by members of the editorial board and by several independent reviewers. In light of the reviews (below this email), we would like to invite the resubmission of a significantly-revised version that takes into account the reviewers' comments.

One point is particularly critical, namely the test of the main claims of the paper with proper stimuli. 

We cannot make any decision about publication until we have seen the revised manuscript and your response to the reviewers' comments. Your revised manuscript is also likely to be sent to reviewers for further evaluation.

Sincerely,

Daniele Marinazzo

Deputy Editor

PLOS Computational Biology

Daniele Marinazzo

Deputy Editor

PLOS Computational Biology

Reviewer's Responses to Questions

**Comments to the Authors:**

Reviewer #1: This manuscript presents an initial and exploratory look at DNNs as models of viscosity perception. The authors join a growing number of researchers in the field in trying to model perceptual computation using DNNs. The authors collected human judgments of viscosity in response to short video clips of simulated liquids. They chose a single DNN architecture to train on the same task and compared its errors to human errors, finding that it accounted for a reasonable amount of the variance. Next, the authors conducted a series of analyses to understand various aspects of the solution(s) found by the model. They conducted representational-similarity analyses to compare unit activations to various hand-engineered features in order to understand the kinds of information that the network uses. As is par for the course when training DNNs as models of perception, the principle conclusion seems to be that neural networks learn complex, difficult-to-interpret functions that don’t map nicely onto any known set of hand-engineered features, but they are currently the best models we have (or at least the best image-computable models), so more work needs to be done. The manuscript represents a promising start, but needs some improvements to the writing, and might be stronger and more interesting with a few additional analyses.

My critiques fall into a few categories:

1. In various places, the interpretation of the network analysis results seemed to be on shaky ground.

i) For example, starting line 474:

“This is further corroborated by the good transfer learning performance in which the final layers were retrained for a different task. It supports the idea that earlier layers converge on task-independent image representations: a basic toolbox of filters that can be applied for a wide range of visual tasks and are similar to the earlier processing stages in our visual cortex.” Are the authors trying to state that a general property of DNNs is to get some amount of task-generalization for free, and therefore this validates DNNs generally as models of perception? If so, this argument needs a little work.

I might recommend something like the following framing to improve flow: The DNNs we trained can perform viscosity judgments about as well as humans, and they are argued to be somewhat plausible models of the perceptual system. Therefore, we reasoned that we could explore the networks’ learned representations to gain some insight into the realm of possibilities for analogous computations in the brain. To the extent that human’s representations are learned wrt similar objectives and constrained in similar ways to the DNNs we trained on this task, our analyses suggest it is unlikely that human viscosity judgments rely exclusively on a set of easy-to-identify features or cues.

The authors might also consider more closely connecting their own work to similar work in other areas of perception (see e.g. Bates & Jacobs (2019) for one review of DNNs as models of behavioral data). I see a handful things are cited, but from the text, I didn’t get a strong impression that this work is well-situated within previous related works.

ii) Starting line 196: “this could suggest that our visual system make use only of the most salient cues in the dataset and ignores the subtler cues that are specific to the training set, which the network learns with greater training”

Or in other words, early learning has better out-of-sample generalization? This strikes me as a rather broad claim about DNNs, so I’d be surprised if there is no literature on it (e.g. in the field of ‘multi-task learning’).

2. Some additional analyses may be warranted to better support the arguments presented, and the motivations for some analyses were unclear.

i) Why not compare other kinds of networks, trained differently or with different architectures? By exploring the space of DNNs a little, we might glean more insight into what elements are essential for capturing human errors. Is there anything specific to the chosen architecture that is important for capturing human errors? E.g., the authors might compare a network that is trained on still images, rather than video, as this eliminates temporal statistics. Are temporal statistics crucial to capture human judgments/errors?

ii) The manuscript presented analyses of how well the hand-engineered features predicted network activations, but I was expecting there to also be some analysis of how well those same features predicted human responses. This seems like an easy sanity check to support the claim that human judgments of viscosity aren’t easy to describe with known features. One could do a regression analysis, or perhaps RDM. Perhaps this was omitted because previous publications have already established it? Or some other obvious reason I’m missing?

iii) From line: 413: “To test this, we measured the ability of the network to classify the different scene classes by applying transfer learning on FC4 and subsequent layers”. Why didn’t you also test the 15-unit network to verify that it couldn’t learn?

But perhaps more importantly, the purpose of the whole enterprise (squeezing FC4) was unclear to me. Regarding this set of analyses, the authors say: “This demonstrates that when networks have capacity that exceeds the bare minimum required for the task, they may encode (i.e., retain, or fail to exclude) aspects of the stimuli that are not strictly necessary for task performance.” Isn’t this just a general property of DNNs? What does it tell us about people’s biases or abilities?

iv) How much does the Bayesian optimization of the hyperparameters help? This procedure was mentioned but I saw no discussion of how much it mattered for the results.

v) Line 239: “This means that the same stimuli are encoded with different ‘mid-level’ image features”.

The interpretation of why the second layer diverges on different re-trainings seems a little shaky, as they only tried out one network architecture, and only looked at FC layers, not conv layers. The relevance of this point to the main findings is also unclear to me. At any rate, I might suggest the authors check out Kornblith, et. al (2019). It might be interesting to see what results from applying the new similarity metric proposed in that paper (which is easy to implement).

vi) Line 281: “To get a clearer impression of the unit-specific function we visualized the stimuli that minimally and maximally activate the unit”. Why not also use gradient methods to find series of images that maximally activate the unit? Perhaps this would provide deeper insights. The authors dismissed this possibility in the discussion, but it wasn’t clear whether they tried it or if they had sound reason to dismiss it a priori as unlikely to be useful. Did reference 22 specifically suggest for their architecture type that it is not useful?

vii) When the network was trained for greater than 30 epochs, on what kinds of simulations did the network diverge from human error? Can anything being discerned by visual inspection or other analysis?

3. Misc.

i) In order to induce errors in participants, why didn’t the authors choose to make the viscosity scale more fine-grained rather than make the images impoverished? I’m slightly concerned that this is less ecological, and therefore the results might not generalize as well. But maybe this is okay because it’s like viewing it from farther away.

ii) Line 212: “This overcomes the challenge of having sufficient labelled data to train directly on human judgments.”

But training on human judgments would result in an empirical (“curve-fitting”) model, which is an entirely different class of model with different associated goals. Therefore this statement may be a little misleading.

iii) Why does Fig. 4 only use 26 networks, instead of 100?

iv) Line 218: “In order to interpret the response patterns, we compare these neural activations with a set of predictors.” —> “In order to interpret the response patterns of individual units, …”

v) Typo, Line 290: “(D) A selection of the RSA correlations for the units closest to the centres of four of the clusters shown in Figure 6: (E)”. Should be Figure 7?

References:

Kornblith, S., Norouzi, M., Lee, H., & Hinton, G. (2019). Similarity of neural network representations revisited. arXiv preprint arXiv:1905.00414.

Jacobs, R. A., & Bates, C. J. (2019). Comparing the Visual Representations and Performance of Humans and Deep Neural Networks. Current Directions in Psychological Science, 28(1), 34-39.

Reviewer #2: Humans are extraordinarily good at making complex inferences about the world based on limited sensory information. One such remarkable ability is to infer intuitive physical properties such as the viscosity of flowing liquids. A model of how humans might use sensory information to make complex inferences has the potential to shed light on how humans might perform this same task, and this is the focus of this particular paper by van Assen and colleagues. To do this, the authors take the approach of training deep convolutional neural networks (DNNs) on a viscosity judgement task and then characterizing internals of this network using representational similarity analysis (RSA) and using other computer vision based descriptors like GIST etc.

The main claim of the paper is: 1) A particular DNN that can predict the behavior of human subjects. The secondary characterization of this DNN revealed units sensitive to certain to specific feature types apart from only viscosity and that it was possible to train a smaller network that achieves similar performance.

The main question raised in the paper is novel and of broad interest. However, the paper suffers several key conceptual issues with the network training paradigm, methods of comparing representations in deep networks and most strikingly, the lack of an alternative DNN model frameworks and an over-reliance on one specific method. Because of these issues, I am unable to recommend the paper for submission in its present form. I synthesize the four central issues with the paper below.

Main Issues:

1. The authors sample from a generative space to train networks which is a reasonable effort given the massive search space of possible videos. However, the authors do not seem to have fully utilized the strength of having this generative space. In particular, from the description in the Methods (Lines 579-580), the authors seem to have used many of the same video stimuli conditions to validate the models during training and to define an early-stopping criterion. Is this really correct? The reason this is confusing and inconsistent is because of Lines 144-147, and then the Lines 152 (“chose network 78 of 100”). What is the criterion of choosing a model and the criterion of defining a premature stop to the training. If the criterion is the match to human behavior, then it amounts to double-dipping, given that the central claim of the paper is the match of the DNN to behavior. This problem can be allayed by training the model (for example) on Scenes 1-8 (from Figure 1) but testing the match to behavior based on Scenes 9-10. The significant concerns about the training procedure and the potential double-dipping issue casts doubt on the authors’ main claims.

2. The authors rely heavily on one and only one model architecture in this paper. There are many DNN models now that can now be trained on videos and could therefore potentially be adapted to this particular task. While I don’t expect the authors to test ALL possible video based models out there, I do think that having at least a couple of additional models is necessary for any claim about this particular DNN model. Or do the authors think that there are many DNN models perform the same task. The writing and framework of the paper suggests one particular DNN framework as a model for this ability. Also, how constrained is the space of models given only one metric of matching behavior?

3. How to compare DNN and other representations? This is an important question and while there are many methods, it has recently been brought to notice that representational similarity analysis (used in the paper) has particular flaws that make it less suitable for this purpose. I would urge the authors to revisit this question using updated tools (eg. Centered kernel Analysis) from statistical physics which have been used to probe this exact question in deep networks. See here for more details:

Similarity of Neural Network Representations Revisited (Korblith et al., 2019) https://arxiv.org/abs/1905.00414

4. From the analyses in Figures 5-7, it seems that the networks extract several low-level spatial and temporal statistics of the videos. While this is interesting, it undermines the argument that viscosity detection is a complex task that does not rely on low-level cues. Another way to look at this would be to train linear decoders from specific layers to extract the viscosity levels. Have the reviewers tried this? How good are early layers of the network at performing the same task?

**Have all data underlying the figures and results presented in the manuscript been provided?**

Reviewer #1: Yes

Reviewer #2: Yes

PLOS authors have the option to publish the peer review history of their article (what does this mean?). If published, this will include your full peer review and any attached files.

Reviewer #1: Yes: Christopher J. Bates

Reviewer #2: No
---

## [Decision Letter · Decision Letter 1]

22 May 2020

Dear Dr. van Assen,

Thank you very much for submitting your manuscript "Visual perception of liquids: insights from deep neural networks" for consideration at PLOS Computational Biology. 

We appreciated the changes to the manuscript, which now looks more robust and convincing. Still a little effort is needed to properly communicate your results and explain their relevance.

Sincerely,

Daniele Marinazzo

Deputy Editor

PLOS Computational Biology

Daniele Marinazzo

Deputy Editor

PLOS Computational Biology

[LINK]

Reviewer's Responses to Questions

**Comments to the Authors:**

Reviewer #1: The authors made various additions in response to comments. The most consequential of these was to train additional network architectures on the task to compare to the original choice. I think this addition has improved the manuscript. I also think the multiple linear regression analysis between the hand-picked features and viscosity was helpful, and should be presented more prominently, as it supports the authors' central argument. Similarly, I think the linear decoding of viscosity from earlier layers provided good results that bolster the central argument, and yet this felt buried.

However, I still find the authors' interpretation of the crucial role for early stopping to be highly speculative, and hand-wave-y. For example, what is precisely meant by "salient cues"? How could this be operationalized in a technical way? There are other possibilities to think about. For instance, if the network were somehow trained on a larger dataset that is more ecological, would we find that it could be trained to convergence and still predict human responses well? In other words, is the choice of dataset driving the early stopping result, or is it some suboptimality or information processing capacity limit in humans? I would encourage the authors to either remove the speculation entirely, or flesh it out more. (I didn't find the brief discussion involving refs 32 and 33 to be very convincing.)

In general, I would also encourage the authors to spend a little more time with the writing in order to increase the impact of the work. The manuscript seems to lose focus and structure at times, which makes it hard to follow and thoroughly evaluate. To me, the most solid takeaway is that the model finds non-trivial features that support viscosity judgment, which are hard to intuit but essential both for doing the task and explaining human judgments. But this point came through a little muddled.

A misc. comment: I didn't see anywhere detailing the methods for "activation maximization" (or even so much as defining what it was).

Reviewer #2: In the paper van Assen and colleagues train deep convolutional neural networks (DNNs) based models on a viscosity judgement task and characterize the internals of this network using representational similarity analysis (RSA) and other computer vision based descriptors like GIST.

I raised some important issues related to the exact way the models were trained (and concerns about potential double-dipping) and over-reliance on a single model architecture for their inference.

The authors have now addressed these issues by clarifying the model training regimen, and by testing a couple other leading video based models. These results, I feel, substantively improve the manuscript. I suggest that the authors emphasize that the models were trained on physical rather than perceived viscosity to make sure that readers do not miss this point.

Going further, the authors also apply more statistically complex methods to compare network representations and show that low-level cues extracted from the videos by themselves not sufficient. This directly addresses the point about low-level features.

Taken together, I feel that the authors have satisfactorily addressed my concerns. I congratulate the authors on their interesting finding and recommend publication in its current form.

**Have all data underlying the figures and results presented in the manuscript been provided?**

Reviewer #1: Yes

Reviewer #2: Yes

PLOS authors have the option to publish the peer review history of their article (what does this mean?). If published, this will include your full peer review and any attached files.

Reviewer #1: No

Reviewer #2: No
---

## [Editor Report · Decision Letter 2]

5 Jun 2020

Dear Dr. van Assen,

We are pleased to inform you that your manuscript 'Visual perception of liquids: insights from deep neural networks' has been provisionally accepted for publication in PLOS Computational Biology.

Best regards,

Daniele Marinazzo

Deputy Editor

PLOS Computational Biology

Daniele Marinazzo

Deputy Editor

PLOS Computational Biology

---

## [Editor Report · Acceptance letter]

30 Jul 2020

PCOMPBIOL-D-19-01964R2 

Visual perception of liquids: insights from deep neural networks

Dear Dr van Assen,

I am pleased to inform you that your manuscript has been formally accepted for publication in PLOS Computational Biology. Your manuscript is now with our production department and you will be notified of the publication date in due course.

With kind regards,

Laura Mallard
